# Quality of Phosphate Rocks from Various Deposits Used in Wet Phosphoric Acid and P-Fertilizer Production

**DOI:** 10.3390/ma16020793

**Published:** 2023-01-13

**Authors:** Urszula Ryszko, Piotr Rusek, Dorota Kołodyńska

**Affiliations:** 1Analytical Laboratory, Łukasiewicz Research Network–New Chemical Syntheses Institute, al. Tysiąclecia Państwa Polskiego 13a, 24-110 Puławy, Poland; 2Fertilizers Research Group, Łukasiewicz Research Network–New Chemical Syntheses Institute, al. Tysiąclecia Państwa Polskiego 13a, 24-110 Puławy, Poland; 3Department of Inorganic Chemistry, Faculty of Chemistry, Maria Curie-Skłodowska University, M. Curie Skłodowska Sq. 2, 20-031 Lublin, Poland

**Keywords:** phosphate rock, wet phosphoric acid, phosphate fertilizers, elemental analysis

## Abstract

Phosphate rocks (PRs) play a crucial role in ensuring the availability of phosphorous for the world’s food needs. PRs are used to manufacture phosphoric acid in the wet process as well as P-fertilizers. The chemical and mineralogical compositions of PRs from Djebel Onk (Algeria), Khneifiss (Syria), Negev (Israel), Bou Craa (Morocco), and Khouribga (Morocco) are discussed in this study. PRs were characterized by inductively coupled plasma optical emission spectrometry (ICP-OES), cold vapor atomic absorption spectrometry (CVAAS), ion chromatography (IC), and X-ray diffraction (XRD), as well as gravimetric and potentiometric methods. All PRs were mainly composed of CaO, P_2_O_5_, SiO_2_, F, SO_3_, Na_2_O, MgO, Al_2_O_3_, Fe_2_O_3_, SrO, and K_2_O at the level of wt.%. The P_2_O_5_ content accounted for 28.7–31.2%, which indicates that these are beneficial rocks to a marketable product. The degree of PR purity expressed by the minor elements ratio index (MER index) varied from 2.46% to 10.4%, and the CaO/P_2_O_5_ weight ratio from 1.6 to 1.9. In addition, the occurrence of trace elements such as As, Cd, Cr, Cu, Hg, Mn, Ni, Pb, Ti, V, U, and Zn, as well as Cr(VI) and Cl ions at the level of mg∙kg^−1^ was found. Since PRs will be used to produce P-fertilizers, their composition was compared with the regulatory parameters set up by EU Regulation 2019/1009 related to the heavy metals (As, Cd, Pb, Ni, Hg, Cu, Zn) and Cr(VI) contents in inorganic fertilizers. The heavy metals and Cr(VI) content in all PRs did not exceed the limit values. XRD analysis revealed that fluorapatite, hydroxyapatite, carbonate fluorapatite, and carbonate hydroxyapatite were the dominant minerals. The accuracy and precision of the used methods were evaluated by analysis of standard reference materials (SRM) for Western Phosphate Rock (NIST 694). The recovery was 85.3% for U and 109% for K_2_O, and the RSD ranged from 0.67% to 12.8%.

## 1. Introduction

Phosphate rock (PR) is a naturally occurring ore of sedimentary or igneous origin that has a high concentration of phosphate minerals [1,2]. This generally accepted term comprises both the unprocessed phosphate ore as well as the concentrated phosphate products [3]. The largest sedimentary deposits are found in northern Africa, the Middle East, China, and the United States. Significant igneous occurrences are found in Brazil, Canada, Finland, Russia, and South Africa. Large phosphate resources have been identified on the continental shelves and seamounts in the Atlantic Ocean and the Pacific Ocean [4].

PRs are mainly mined and processed to form phosphoric acid as well as P-fertilizers, both single- and multi-component. PRs are nonrenewable resources and have no substitutes in agriculture. The global demand for PRs is steadily increasing due to an ever-growing population and, thus, an unrelenting food supply. The production increase is directly linked to the availability of P-fertilizers and, consequently, PRs [5,6]. Figure 1 shows the world’s growth in phosphate rock production.

In 2021, the worldwide PRs mine production was 220 million tons, and the reserves were 71 billion tons. The most important producers are China, Morocco, the United States, and Russia accounting for over 70% of the total exploitation of deposits [4] (Figure 2). Europe is almost completely excluded from access to this wealth, except for Russia and small deposits in Finland [7]. The European Union relies on nearly 90% of PRs, which is why the European Commission considers this raw material to be critical for the European economy [8].

The estimates of world production and reserves for the major worldwide producers are given in Table 1. Individual phosphate rock deposits have been exploited for a long time with varying intensity; therefore, the distribution of the reserves does not entirely coincide with the area of their exploitation. Some resources are inaccessible or uneconomical, others are economical but require underground mining, and still, others require only the removal of a relatively shallow overburden [6].

The production of P-fertilizers requires an intermediate step: the production of phosphoric acid, which is one of the most important intermediates of the inorganic chemical industry. Phosphoric acid is manufactured from phosphate ores using thermal and wet processes [9]. In industry, the wet process is more commonly used due to its lower energy requirements compared to the thermal process and, thus, greater economic viability [10].

In Europe, the most common method for the production of wet phosphoric acid (WPA) is the dihydrate process (DH). WPA is manufactured by the reaction of the phosphate rock with concentrated sulfuric acid in the temperature range of 70–80 °C, according to the simplified reaction:[3Ca_3_(PO_4_)_2_]·CaF_2_ + 10H_2_SO_4_ + 20H_2_O → 6H_3_PO_4_ + 10CaSO_4_·2H_2_O + 2HF

The insoluble calcium sulfate is then separated from the phosphoric acid, most usually by filtration. In this stage, the acid concentration is approx. 26–30% P_2_O_5_. For use in fertilizer production or further processing, the acid is concentrated in two steps to 40% and approx. 50% P_2_O_5_ [9,11]. The acid can be clarified after each of the concentration steps. The manufactured WPA is further processed to produce a specific type of P-fertilizer, as shown in Figure 3.

The production of phosphoric acid by the wet process frequently gives a product that contains impurities [11]. The type of contaminants in WPA depends on the chemical composition and physical properties of the PR, the type of sulfuric acid (contact, metallurgical, waste), and applied technology [12]. Contaminants move to various degrees from PR to WPA and phosphogypsum [13]. Mineral and organic impurities of PR determine its suitability for the production of WPA. The origin of PR is thus of critical importance since some areas are reputed for the presence of high concentrations of hazardous metal ions that substantially decrease the commercial value of produced WPA. The main impurities of WPA are the compounds of Ca, Al, Fe, Mg, Na, and K, as well as F and Cl ions. WPA also contains heavy metal ions such as Cd, As, Hg, Pb, Cr, and Ni, as well as silica, U, organic substances, carbon, and lanthanide present at the level mg∙kg^−1^ [12]. An appropriate purity level of phosphoric acid is required for particular applications [14].

Phosphate deposits vary in composition around the world and even within a local mine, hence the need for regular analysis of the mined raw materials [6]. The level of impurities in the PR should be such as to enable the production of P-fertilizers in accordance with the applicable law.

The use of P-fertilizers in agriculture is framed by Regulation EU 2019/1009 of 5 June 2019, laying down the rules for making fertilizing products available on the EU markets, which determine permissible concentrations of heavy metals in mineral fertilizers to prevent environmental pollution and to ensure safety for human health. The regulation came into force on 16 July 2022 and banned the sale of inorganic fertilizers containing specific toxic contaminants, including lead (Pb), mercury (Hg), hexavalent chromium (Cr(IV)), nickel (Ni), arsenic (As), and cadmium (Cd), as well as zinc (Zn) and copper (Cu) in the amounts exceeding the set of maximum concentration values specified in the regulation 2019/1009 [15].

The aim of this study was to compare the composition of phosphate raw materials from various deposits in order to assess their suitability for WPA and P-fertilizers production. PRs quality assessment was based on P_2_O_5_ content and chemical and mineralogical composition. Contaminants of PR determine its suitability for WPA manufacture and decrease its commercial value. Quality control of raw materials is an important part of the production process, as they should fulfill certain quality requirements. The suitability of all tested PRs for the WPA and P-fertilizers production has been confirmed.

## 2. Materials and Methods

All used reagents were analytical grade. The solutions were prepared with deionized water obtained from the water purification system (Milli-Q, Merck, Darmstadt, Germany). The nitric, hydrochloric, and fluoric acids were of suprapure quality (Merck, Germany). The glassware used in the experiments was previously decontaminated with an HNO_3_ solution (10% v∙v^−1^) for 24 h, subsequently washed with ultrapure water, and dried at room temperature.

### 2.1. Phosphate Rocks Samples (PRs)

The samples denoted as PR1, PR2, PR3, PR4, and PR5 were obtained from the mines Djebel Onk (Algiers), Khneifiss (Syria), Negev (Israel), Bou Craa (Morocco), and Khouribga (Morocco), respectively. It should be noted that these are beneficial rocks, that is, mined phosphate ore that may have been washed and separated from the gangue by flotation, sometimes calcined, sized, and concentrated up to a marketable product. Standard reference material (SRM) for Western Phosphate Rock (NIST 694) was employed to assess the quality of the results obtained by applying the proposed analytical methods. This material presents a composition similar to that of the phosphate rocks samples and is adequate as a reference material.

### 2.2. Analytical Procedures

#### 2.2.1. Elemental Analysis by ICP-OES

For the simultaneous determination of the total major elements Ca, Si, Na, Mg, Fe, Al, K, and Sr, as well as the trace elements such as Cd, Mn, Cu, Zn, Ti, V, Cr, Ni, U, As, and Pb, the inductively coupled plasma optical emission spectrometer with the axial view (ICP-OES, 720-ES, Varian, Mulgrave, Australia) was applied. All test samples, blanks, and calibration solutions were measured under the same optimized conditions detailed in Table 2. It should be mentioned that the Si analysis by ICP-OES requires using a suitable sample introduction system consisting of a Teflon cyclonic spray chamber, a Vee Spray ceramic nebulizer, and a demountable torch with a ceramic inner tube.

The ICP-OES technique requires the complete transfer of the sample into the solution using mineralization or dissolution of samples before the analysis. Sample preparation is the most challenging step in trace analysis as well as the most significant source of error. PR samples microwave digestion was conducted in the closed system (Mars, CEM, USA), using the mixture of 65% HNO_3_ and 37% HCl at the volume ratio of 3:1. The combination of HNO_3_ with HCl in a closed system increases the efficiency of digestion due to the high oxidizing power of the mixture and the higher pressure and temperature generated inside the digestion vessel [16]. To prepare the PRs samples for Si determination, it was necessary to carry out total mineralization with the mixture of HF, HNO_3_, and HCl acids. The microwave digestion parameters were based on the available literature data and earlier experiments. All procedures were performed in triplicate, including the blank solution and SRM, and then the average value was set.

#### 2.2.2. P and S Determination by Gravimetric Methods

The total phosphorus content in conversion to P_2_O_5_ was determined according to the Polish Committee of the Standardization standard, PN-EN 15956:2011 and PN-EN 15959:2011. These methods involve the extraction of phosphorous contained in the phosphate rocks using a mixture of HNO_3_ and H_2_SO_4_. Extracted phosphorous is determined with the gravimetric method using quinoline phosphomolybdate (after possible hydrolysis). After filtering and washing, the sediment is dried at the temperature of 250 °C and weighed. Under these conditions, the compounds contained in this solution (mineral and organic acids, ammonium ions, soluble silicates) do not hinder the determination process if sodium molybdate or ammonium molybdate reagents are used for precipitation.

The total sulfur determination in conversion to SO_3_ was based on the PN-EN 15925:2011 and PN-EN 15749:2012 (method A) standards.

#### 2.2.3. Hg Determination by CVAAS

For the total Hg analysis, the Mercury Analyzer (DMA-80 evo, Milestone, Italy) was applied. The method is based on the direct amalgamation technique without previous digestion and involves the thermal decomposition of the sample in an oxygen-rich environment. The decomposition products were transferred to an amalgamator, which selectively captures Hg. After flushing the system with oxygen to remove any remaining gas or decomposition products, the amalgamator heats up rapidly, releasing Hg vapor. The absorbance was measured at the 253.7 nm wavelength as a function of the Hg concentration in the sample.

#### 2.2.4. Cr(VI) Determination by IC-DPC

For the extraction of Cr(VI) content from the PRs samples, the alkaline digestion procedure was applied. This consisted of 1.25 g of the sample, weighed into a 50 mL digestion vessel, and with the 25 mL extraction solution containing 0.5 mol∙L^−^^1^ NaOH and 0.28 mol∙L^−^^1^ Na_2_CO_3_ added. Then, 1 mL of 4 mol∙L^−^^1^ MgCl_2_ was added to prevent oxidation of Cr(III) during the extraction procedure, and 0.5 mL buffer solution of each 0.5 mol∙L^−^^1^ K_2_HPO_4_ and KH_2_PO_4_. The digestion vessel was capped with a watch glass, transferred to the heating block, and its contents were heated for 1 h at 92.5 °C. After the extraction, the samples were cooled to room temperature, filled up the solution to 50 mL with the ultrapure water from the Milli-Q water system, and filtered through the 0.45 mm membrane filter.

For the selective determination of Cr(VI) as CrO_4_^2−^ ion, the ion chromatography technique with the UV-VIS detection based on the ICS-3000 high-performance ion chromatograph system from Dionex (Sunnyvale, CA, USA) was used. The system is equipped with the eluent reservoir, isocratic pump, valve, post-column reaction reagent feeder (PC-10), reaction loop (Dionex Reaction coil, 375 μL), and a dual-beam UV/VIS detector (AD-25) with the deuterium and tungsten lamp as well as the readable values for λ = 540 nm. After elution, the Cr(VI)-complex was formed using the post-column derivatization reaction with 1.5-diphenylcarbazide (DPC) through the reaction coil [17]. The separation conditions are given in Table 3.

#### 2.2.5. F analysis by the Potentiometric Method

The fluoride selective electrode, Monokrystaly 09–37 type, made of lanthanum fluoride single crystal (LaF_3_), was used as the indicating electrode combined with the AgCl electrode used as the reference one. The citrate buffer was used to maintain the required pH value demanding the linear relationship between the measured signal and log_10_ of the molar concentration of standard fluoride solutions. The standard fluoride solution of 1000 mg F∙L^−^^1^ was prepared by dissolving 2.21 g sodium fluoride (Acros Organics) (dried at 80 °C) in the 1000 mL volumetric flask and making the volume up to mark with ultrapure water from the Milli-Q water system. The experimental solutions of desired concentrations were prepared by diluting the standard solution. 

#### 2.2.6. Cl Analysis by the IC Method

The chloride determination by the ion chromatography method was based on the PN-EN ISO 10304-1:2009 standard.

#### 2.2.7. Mineralogical Characteristics of Phosphate Rocks by the X-Ray Diffraction

The XRD measurements were made using a powder diffractometer operating in the Bragg-Brentano geometry (Empyrean, Panalytical, Malvern, UK). The standard equipment includes a Cu X-ray tube, a rotating transmission-reflection table, and a fast linear PIXcel 3D detector. The intensity and voltage of the X-ray tube were I = 40 mA and U = 40 kV, respectively. The PreFix system configuration (pre-aligned fast interchangeable X-ray modules) made it possible to carry out measurements with the settings given in Table 4.

Dimensions of the test substance disc: diameter 16.2 mm, depth 2.2 mm. Specimen support was used to allow rear charging, as no increased preferred orientation of the tested materials was found. The tests for all samples were measured using the same program with the same incident beam and diffracted beam optics. The diffractogram analysis was performed in the HighScorePlus program.

Due to the requirements in the sample preparation process for the XRD measurements, the materials were considered homogeneous and were only sieved to obtain a fraction below 0.071 mm. Additionally, the PR1 sample was dried for 12 h at 105 °C and prepared for the measurement in the side-loading table to determine the influence of the preferred orientation and humidity on the obtained diffraction spectra. No influence of additional thermal treatment and crystallite shape on the obtained results was found. The nature of all samples was similar, so PR1 was treated as a representative sample for the evaluation of these changes.

## 3. Results and Discussion

### 3.1. The Elemental Composition of Phosphate Rocks

The elemental analysis results of PRs showed that they were mainly composed of CaO, P_2_O_5_, SiO_2_, F, SO_3_, and Na_2_O, as well as MgO, Al_2_O_3_, Fe_2_O_3_, SrO, and K_2_O. The CaO content accounted for 48.4–57.1%. The content of P_2_O_5_ in PR1, PR2, PR3, PR4, and PR5 was 29.0, 28.7, 30.4, 29.4, and 31.3%, respectively. The smallest concentration of P_2_O_5_ was found in PR2, and the largest was in PR5.

The concentration of P_2_O_5_ in the ore determines the possibility of its processing; therefore, the deposits can be described as rich or poor in phosphorus. Based on the P_2_O_5_ content, three phosphate grades can be distinguished: low-grade ores (12–16% P_2_O_5_), medium-grade ores (17–25% P_2_O_5_), and high-grade ores (26–35% P_2_O_5_). The phosphate ore enrichment processes allow the gangue minerals to be separated from the economical phosphate value [18]. Several methods are used to upgrade the low-grade phosphate ore to a marketable-grade product, such as crushing, grinding, screening, scrubbing, heavy media separation, washing, roasting, calcination, and flotation. Phosphate deposits, which after mining and processing, yield a raw material containing 28–38% P_2_O_5_, are considered economically viable [19].

There are many considerations when selecting a PR supply for the production of wet-process acid. One of the factors taken into account in assessing the quality of deposits is the CaO/P_2_O_5_ weight ratio, as this governs the amount of sulfuric acid that is required to acidulate the PR in WPA manufacture. The CaO/P_2_O_5_ weight ratio is a common measure of phosphate rock quality. For pure apatite, the ratio is 1.32, and the commercially available rock may have a CaO/P_2_O_5_ ratio of up to 1.6 [6]. In the analyzed PRs samples, the CaO/P_2_O_5_ ratio accounted for 1.6–1.9.

The content of major constituents of PRs, such as Al_2_O_3_, K_2_O, MgO, Na_2_O, SO_3,_ and SrO, varied depending on their origin and accounted for 0.13–0.51% Al_2_O_3_, 0.10–0.22% K_2_O, 0.34–1.31% MgO, 0.52–1.54% Na_2_O, 1.46–3.36% SO_3_, and 0.12–0.27% SrO. Those results show no significant differences in the PRs composition, contrary to the content of SiO_2_ and Fe_2_O_3_. A relatively high concentration of SiO_2_ was found in the PR2 (8.67%) and PR5 (5.77%) samples compared to PR1 (2.93%), PR3 (1.67%), and PR4 (2.66%). The content of Fe_2_O_3_ in PRs varied from 0.205 to 0.64%, except for the PR5 deposit, in which the concentration of Fe_2_O_3_ was found to be higher (2.31%).

The Al, Fe, and Mg compounds are the main impurities that move from PRs to phosphoric acid in the wet process [19]. The impurities impart undesired color, turbidity, and viscosity and can increase the corrosiveness of WPA [20]. An increase in the Al content causes an increase in the density and viscosity of WPA, making it difficult to process it. In the concentrated WPA, it can precipitate in the form of orthophosphates(V). Nevertheless, the presence of Al has a positive effect on acid filtration. The Fe, and likewise the Al, in the concentrated acids can precipitate in the form of orthophosphates(V), causing P_2_O_5_ losses. The large Fe content affects the viscosity of the WPA negatively. The presence of Mg in WPA is undesirable due to its effect on the increase in viscosity and the formation of insoluble precipitates [6,21,22].

In industrial practice, to determine the degree of purity of the raw material, the MER index (*minor elements ratio*) is typically used, which is the sum of the oxides of the major impurities relative to the P_2_O_5_ content:MER(%) = (Al_2_O_3_ + Fe_2_O_3_ + MgO/P_2_O_5_) × 100(1)
where Al_2_O_3_, Fe_2_O_3_, MgO and P_2_O_5_ are expressed in wt.%.

The value of the MER index in PR1, PR2, PR3, PR4, and PR5 was 8.46%, 3.04%, 2.46%, 6.82%, and 10.4%, respectively. Low MER is important for the rock to be processed to WPA and P-fertilizers.

Among the main impurities that move from the phosphate raw material to the WPA, Cl and F ions should also be mentioned. The content of Cl in the analyzed PRs was at the level of 100 mg∙kg^−^^1^, except for PR2 (635 mg∙kg^−^^1^) and PR3 (1278 mg∙kg^−^^1^). Even though the process of obtaining WPA is corrosive and requires the use of acid-resistant steel, too high a concentration of Cl ions (above 0.1% by mass) can cause corrosion of this steel [6]. The content of F in the phosphate raw material can be as high as 3–4% by mass [23] which is confirmed in this study. The F content varied from 3.12 to 4.07%. During the extraction process, HF and SiF_4_ pass into the flue gases, from which H_2_SiF_6_ is formed in the absorption process. The silica hexafluoride ion reacts with the Na and K ions precipitate and may hinder the process of exhaust gas purification and filtration [6]. PRs are generally pretreated by calcination and digestion beneficiation. During the calcination and digestion, F and Cl in PRs can be released by volatilization and dissolution [24].

Due to the fact that PRs will be used to produce WPA and P-fertilizers, the composition of PRs was compared with the regulatory parameters set up by EU Regulation 2019/1009 relating to the macronutrient inorganic fertilizers. The regulation sets permissible values for the heavy metal (As, Pb, Cd, Ni, Hg) and Cr(VI) contents depending on the type of fertilizer product. The content of As in PRs varied from 5.24 to 23.8 mg∙kg^−1^, and all PRs samples had As concentrations below the maximum value (40 mg∙kg^−1^). In the case of Pb, all samples also fulfilled the requirement (120 mg∙kg^−1^), and the Pb concentration was below the LOQ value (<8 mg∙kg^−1^) of the ICP-OES method. The content of Hg was much lower than the maximum concentration (1 mg∙kg^−1^) as well as the Cr(VI) content (2 mg∙kg^−1^) at the µg∙kg^−1^ level. The permissible value for the Cd depends on the P content in macronutrient inorganic fertilizer. Where fertilizer has a total P content of less than 5% P_2_O_5_ equivalent by mass, the Cd content must not exceed 3 mg∙kg^−1^, or if it has a P content equivalent or more by mass of 5% P_2_O_5_ (P-fertilizers), the Cd content must not exceed 60 mg∙kg^−1^ P_2_O_5_ [15]. The Cd concentration in PR1, PR2, PR3, PR4, and PR5 was 14.3, 7.52, 24.8, 14.5, and 16.0 mg∙kg^−1^, respectively. The results from this study show a good agreement with the literature reports [2,25].

The EU Regulation 2019/1009 also determines the permissible levels of Cu (600 mg∙kg^−1^) and Zn (1500 mg∙kg^−1^); however, these limit values will not apply where Cu or Zn has been intentionally added to an inorganic fertilizer to correct a soil micronutrient deficiency. Moreover, the Cu and Zn contents in PRs were relatively small, 94–98% and 68–90%, respectively, smaller than the maximum permitted values.

Identification and quantification of key sources of heavy metals and Cr(VI) in the phosphate raw materials and their control are crucial for reducing heavy metal content in the P-fertilizers and reducing contamination releases to the environment through agricultural applications. The results of this study presented in Table 5 show that the As, Cd, Pb, Ni, Hg, Cu, Zn, and Cr(VI) contents in all analyzed PRs did not exceed the limit values.

PRs contain naturally occurring uranium (U), and the radioactive components of the U decay series are associated with the phosphate material [26]. During the digestion of PR by the dihydrate process, most of the U is presented in the phosphoric acid solution. It is estimated that 85–95% of the U in the PR feed goes into the solution [21]. The amount of U varied from 34.7 to 110 mg∙kg^−1^ in analyzed samples. In addition, in the composition of the analyzed PRs, the occurrence of trace elements such as Cr, Mn, Ti, and V at the level of mg∙kg^−1^ was found. 

The analysis of the composition of PRs may not be sufficient to assess whether a given material will be suitable for use in a given plant for WPA and P-fertilizer production. For this reason, before selecting a particular raw material or its mixture, it is necessary to carry out laboratory tests with it as well as pilot production tests (extraction, filtration, concentration), which will allow the selection of appropriate conditions for the technological process.

The precision of the method was evaluated via repeatability and can be expressed as the relative standard deviation (RSD) of a series of measurements. The accuracy expresses the difference between the value found experimentally and the reference value. In this study, Western Phosphate Rock (NIST 694) was used to establish the accuracy through the values calculated using Equation (2).
Recovery (%) = (found value/certified value) × 100(2)
where the *found value* is the analyte concentration determined by the proposed method, and the *certified value* is the concentration value of the analyte reported in the SRM certificate of analysis.

The precision and accuracy expressed as RSD (%) and recovery (%), respectively, obtained for the optimized analytical methods, as well as the certified values and the found values for SRM, can be found in Table 6. The recovery was between 85.3% for U and 109% for K_2_O, and the RSD values ranged from 0.67% to 12.8%.

### 3.2. The Mineralogical Composition of Phosphate Rocks

The content of individual crystalline phases was determined by the semi-quantitative Rietveld method with the crystal structure refinement with the help of the below-mentioned cards from the PDF-4 + database. The total content of apatites was shown compared to the other phases in the samples. The size of the apatite crystallites was not estimated due to the overlapping of many reflections. The width of the reflections in the case of apatites is comparable, so the average size of the crystallites should be similar (Table 7).

In the sample PR1, there are reflections characteristic of the angle 2θ: 32.0°, 32.3°, 33.3°, and 34.2° of the fluorapatite carbonate phase. The fluorapatite carbonate reflections are widened, in particular, their base on their left side, which is easy to observe at the angles 2θ: 25.5°, 28.9°, and 31.8°. This suggests the presence of hydroxyapatite carbonate with the reflections characteristic of the angle 2θ: 31.9°, 32.2°, 33.1°, and 34.1°, as well as hydroxyapatite with the reflexes characteristic of the 2θ angle: 61.3°, 31.8°, 32.2°, and 32.9°. The hydroxyapatite carbonate and hydroxyapatite phases complement the reflections from fluorapatite carbonate, which is the major part of the overlapping reflections. Due to the similar angular reflexes of these phases, they are denoted in the graphic as Ca_5_(PO_4_)_3_[F,OH,CO_3_] as a group of phases with similar characteristic reflections without distinguishing between individual phases of apatite. In addition to the recognized phases of apatite, the sample probably includes the ankerite and dolomite phases, the main reflections of which are visible at the angles of 2θ: 30.8° and 30.9°, respectively. It is possible that the sample also contains calcite, whose reflexes coincide partially with those of apatite and widen them on their right at the angle of 2θ of 29.4°. Another phase that can be attributed to the reflections at the angles of 2θ: 20.8° and 26.5° is quartz. The X-ray powder diffractogram of PR1 is shown in Figure 4a.

Figure 4b shows the diffractogram of PR4, which in terms of diffraction lines, their width, intensity, and thus the content and occurrence of phases is similar to PR1. The main difference can be observed at the angle of 2θ of 30.8°, based on the presence of ankerite and dolomite phases. In the case of PR4, the reflex is lower, and no “double peak” is visible in it, as was the case with PR1, meaning that the dolomite content must be smaller.

Figure 5a shows the XRD pattern of PR2. Compared to the diffraction patterns described previously, the differences are significant, both in terms of the presence of phases and their content. The reflections characteristic of the apatite phase are shifted towards the smaller 2θ angles, which can be the result of a greater proportion of other apatite phases than the fluorapatite carbonate phase. The dominant phase in this sample is probably hydroxyapatite with the reflections characteristic of 2θ angles: 25.8°, 31.9°, 33.2°, and 46.9°, and presumably the complementary phase, increasing the width of hydroxyapatite reflections at the base is probably hydroxyapatite carbonate with the reflections characteristic of the 2θ angles: 31.9°, 32.2°, 33.1°, and 34.1°. Based on the diffractograms, it is not possible to indicate whether it is fluoroapatite in the PR2 sample. This sample has a much larger quartz content with visible reflections at the angles of 2θ: 20.8°, 26.6°, and 50.0°. It is also possible to indicate clearly the presence of calcite, the reflexes of which are located at the angles 2θ: 29.4°, 35.9°, 43.2°, and 48.5°. Probably it occurs here, as before ankerite.

The XRD pattern of PR3 is presented in Figure 5b. Similar to the sample PR2, the main phase is probably hydroxyapatite with the reflections characteristic of the 2θ angles: 25.8°, 31.9°, 33.2°, and 46.9°. As for PR2, it is difficult to indicate other phases, such as hydroxyapatite carbonate, the reflections of which can be mostly “covered” by hydroxyapatite. The phase that can be clearly defined based on the visible and high reflections is calcite. The remaining phases with small contents are probably quartz for the angles 2θ of 20.8° and 26.6°; ankerite for the angle 2θ of 30.8°; dolomite for the angle 2θ of 30.9°; and gypsum for 2θ angles of 11.6°, 20.7°, 23.4°, and 29.1°.

In the sample PR5, the dominant phase is that of fluorapatite or hydroxyapatite. The reflections from these phases have similar intensities and locations, hence the difficulty in the qualitative and quantitative determination. In addition, the sample contains calcite with the characteristic reflections at 29.4°, 35.9°, and 43.1°; and quartz with the principal reflections at 26.6° and 28.8°. Dolomite with the prominent feature reflection at the 2θ of 30.9° is likely to be found. The diffractogram of PR5, depending on the intensity of the signal from the angle 2θ, is shown in Figure 5c.

## 4. Conclusions

Quality control of PRs is of great importance in phosphoric acid production as the efficiency of the wet-process acid plants depends primarily on the rock value. In this study, PRs from various deposits were characterized and compared in terms of the P_2_O_5_ content and undesirable contaminants, the level of which depends on the origin and type of deposit. The P_2_O_5_ concentration is directly related to the amount of WPA produced in the wet-process acid plant, and the CaO content governs the amount of sulfuric acid that is required to acidulate the PR. Moreover, impurities present in the PR influence the physicochemical properties of WPA and the technological parameters for its production. This affects the cost of acid production and plant profitability. The PR impurities, especially heavy metals, determine their suitability for the production of P-fertilizers. P-fertilizers launched onto the EU market should meet strict quality and safety requirements in accordance with the applicable law. 

## Figures and Tables

**Figure 1 materials-16-00793-f001:**
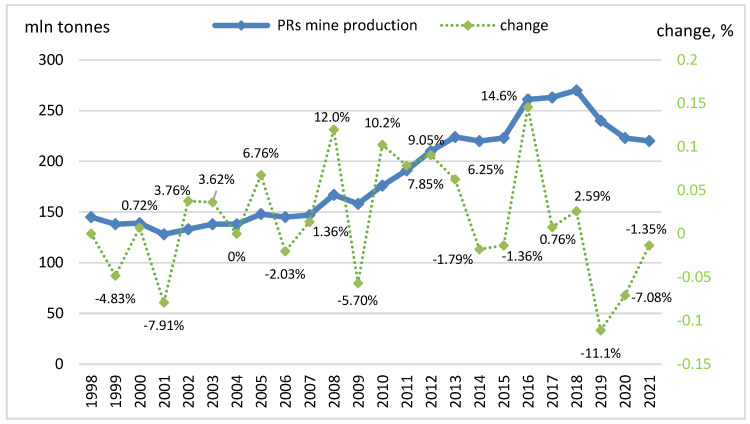
World phosphate rock production in 1998–2021 [4].

**Figure 2 materials-16-00793-f002:**
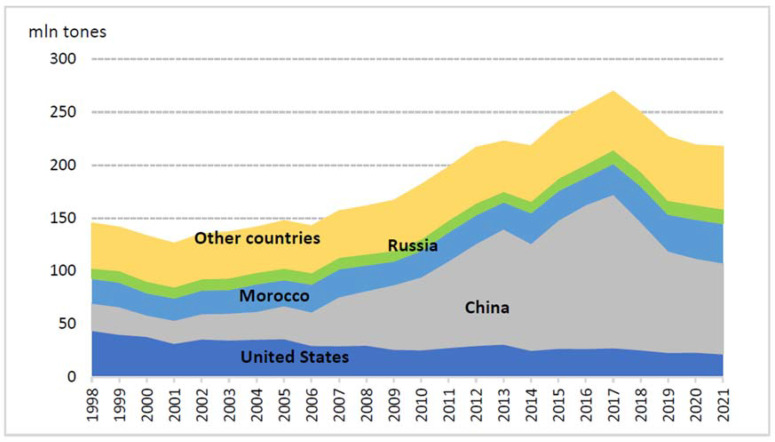
World phosphate rock production in 1998–2021 [4].

**Figure 3 materials-16-00793-f003:**
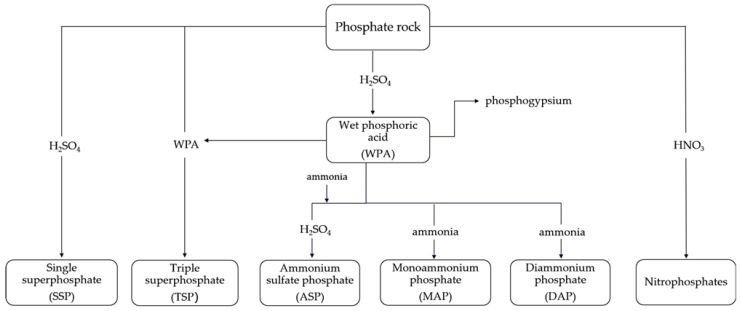
Relationship between the phosphate rock and the phosphate fertilizers [3].

**Figure 4 materials-16-00793-f004:**
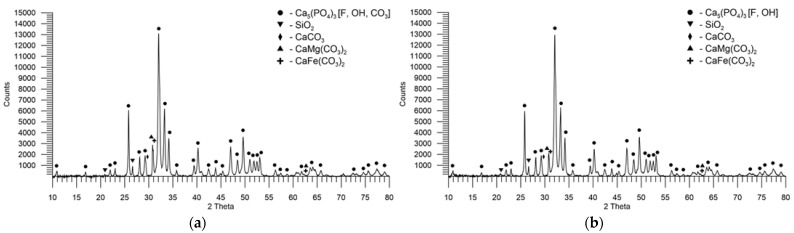
XRD pattern of PRs from various deposits: (**a**) PR1; (**b**) PR4.

**Figure 5 materials-16-00793-f005:**
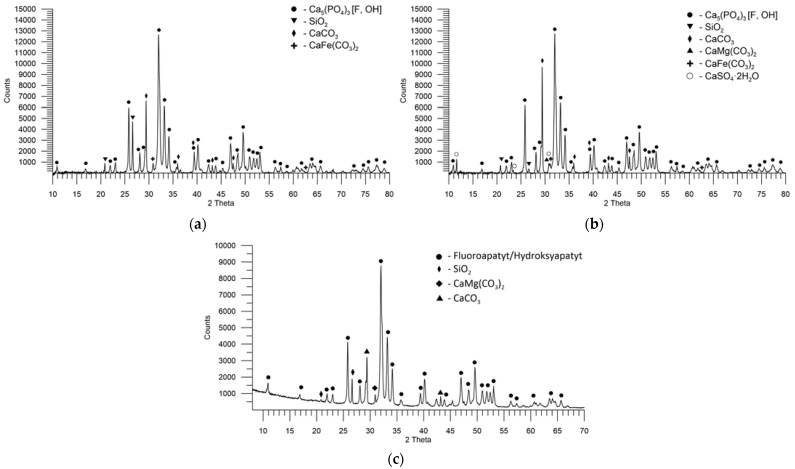
XRD pattern of PRs from various deposits: (**a**) PR2; (**b**) PR3; (**c**) PR5.

**Table 1 materials-16-00793-t001:** World production and reserves of phosphate rocks in 2020–2021 [4].

	Mine Production		Reserves
	2020	2021 ^1^	
	Million Tons		
United States	23.5	22	1000
Algeria	1.2	1.2	2200
Australia	2	2.2	1100 ^2^
Brazil	6	5.5	1600
China ^3^	88	85	3200
Egypt	4.8	5	2800
Finland	0.995	1	1000
India	1.4	1.4	46
Israel	3.09	3	53
Jordan	8.94	9.2	1000
Kazakhstan	1.3	1.5	260
Mexico	0.577	0.53	30
Morocco	37.4	38	50,000
Peru	3.3	3.8	210
Russia	14	14	600
Saudi Arabia	8	8.5	1400
Senegal	1.6	2.2	50
South Africa	1.8	2	1600
Togo	0.942	1.2	30
Tunisia	3.19	3.2	100
Turkey	0.6	0.6	50
Uzbekistan	0.9	0.9	100
Vietnam	4.5	4.7	30
Other countries	0.87	1	2600
World total (rounded)	219	220	71,000

^1^ Estimated; ^2^ For Australia, Joint Ore Reserves Committee-compliant or equivalent reserves were 110 million tons; ^3^ Production data for only large mines, as reported by the National Bureau of Statistics of China.

**Table 2 materials-16-00793-t002:** Characteristics and operating conditions used for the analysis by ICP-OES with the axial view.

Parameter	Characteristics	
Radiofrequency power (W)	1200	
Plasma gas flow rate (L∙min^−1^)	15.0	
Auxiliary gas flow rate (L∙min^−1^)	2.25	
Nebulizer gas flow rate (L∙min^−1^)	0.75	
Nebulizer type	Concentric, conical ^1^	
Spray chamber	Type cyclonic ^1^	
Replicates	3	
Injector tube diameter (mm)	1.8	
Signal integration time (s)	15	
Wavelength (nm)	Al I 396.152	Na I 589.592
	As I 188.980	Ni II 231.604
	Ca II 317.933	Pb II 220.352
	Cd II 214.439	Si I 250.690
	Cr II 267.716	Sr II 216.596
	Cu I 327.595	Ti II 334.188
	Fe II 238.204	U II 409.013
	K I 766.491	V II 311.837
	Mg II 279.553	Zn I 213.857
	Mn II 257.610	

^1^ Excluding Si analysis; (I) Atomic line; (II) Ionic line.

**Table 3 materials-16-00793-t003:** Separation conditions for the ion chromatograph ICS-3000, Dionex [17].

Guard column	IonPac AG7 (4 × 50 mm)
Analytical column	IonPac AS7 (4 × 250 mm)
Eluent	250 mmol∙L^−1^ (NH_4_)_2_SO_4_/100 mmol∙L^−1^ NH_3aq_
Eluent flow rate	1.0 mL∙min^−1^
DPC as PCR flow rate	0.6 mL∙min^−1^
Injection volume	120 μL
Detection	UV/Vis

**Table 4 materials-16-00793-t004:** Optical settings and scan parameters of the powder diffractometer (Empyrean, Panalytical).

Optical Path Parameters for the Incident Beam	
soller gap	0.04 rad
mask	5 mm
beam divergence adjustment slot	½^0^
anti-scatter gap	1^0^
**Optical Path Parameters for the Diffracted Beam**	
anti-scatter gap	8 mm PIXcel 3D
soller gap	0.04 rad
**Scan Parameters**	
initial angle [2θ]	10°
end angle [2θ]	80°
scan step [2θ]	0.0262°
exposure time, s	70.125
sample rotation	disc rotation with the period of 4 s

**Table 5 materials-16-00793-t005:** The concentration of major and trace elements in the phosphate rocks from various origins.

Constituents	PR1	PR2	PR3	PR4	PR5
Major elements concentration, wt.%					
**P_2_O_5_**	29.0	28.7	30.4	29.4	31.2
**CaO**	53.2	53.4	57.1	48.4	55.8
**Al_2_O_3_**	0.51	0.33	0.13	0.44	0.51
**F**	3.12	3.53	3.32	3.47	4.07
**Fe_2_O_3_**	0.64	0.21	0.27	0.55	2.31
**K_2_O**	0.20	0.10	0.22	0.22	0.13
**MgO**	1.31	0.341	0.348	1.01	0.413
**Na_2_O**	1.54	0.61	0.52	1.09	0.6
**SO_3_**	3.36	1.46	2.68	2.71	2.36
**SiO_2_**	2.93	8.67	1.67	2.66	5.77
**SrO**	0.27	0.19	0.25	0.26	0.12
Trace elements concentration, mg∙kg^−1^					
**As**	12.8	5.24	10.8	14.2	23.8
**Cd**	14.3	7.52	24.8	14.5	16.0
**Cl**	117	635	1278	149	136
**Cr_total_**	187	124	77.5	176	217
**Cr(VI)**	0.42	0.58	0.78	0.28	0.28
**Cu**	10.6	26.5	28.9	10.5	37.0
**Hg**	0.020	0.023	0.053	0.020	0.023
**Mn**	72.9	10.7	15.3	70.8	11.7
**Ni**	14.9	23.5	54.4	17.5	30.7
**Pb**	<8.0	<8.0	<8.0	<8.0	<8.0
**Ti**	109	67.2	32.4	84.4	133
**U**	40.5	72.1	110	34.7	106
**V**	63.5	125	99.4	64.3	184
**Zn**	161	302	475	152	263

**Table 6 materials-16-00793-t006:** Analytical results obtained for SRM 694 by the ICP-OES method and their comparison with the certified values.

Constituents	Certified Value ± U, wt. %	Found Value, wt. %	Recovery, %	RSD, % n = 5
Al_2_O_3_	1.8 ± 0.1	1.76	97.7	4.75
CaO	43.8 ± 0.4	44.7	103	5.06
CdO	0.015 ± 0.003	0.015	100	1.60
F ^1^	3.2 ± 0.1	3.19	99.5	0.67
Fe_2_O_3_	0.79 ± 0.06	0.69	87.8	3.72
K_2_O	0.51 ± 0.02	0.56	109	6.45
MgO	0.33 ± 0.02	0.32	97.2	2.62
MnO	0.0116 ± 0.0012	0.0112	96.6	5.22
Na_2_O	0.86 ± 0.04	0.77	89.7	7.29
P_2_O_5_ ^2^	30.2 ± 0.1	30.2	100	-
SiO_2_	11.2 ± 0.4	12.0	107	12.8
U	0.01414 ± 0.00006	0.01206	85.3	6.09
V_2_O_5_	0.31 ± 0.07	0.30	95.9	0.90

^1^ Result obtained by the potentiometric method (n = 2); ^2^ Result obtained by the gravimetric method (n = 1).

**Table 7 materials-16-00793-t007:** The content of individual phases determined by the Rietveld method.

	PR1	PR2	PR3	PR4	PR5
	%				
Apatite ^1^	92.6	82.5	80.8	95.0	91.0
Quartz	0.9	6.1	0.4	1.0	2.3
Dolomite	2.7	1.9	1.5	1.6	1.1
Ankerite	2.9	-	1.2	2.4	
Calcite	0.9	9.6	12.4	-	5.5
Gypsum	-	-	3.7	-	

^1^ Group of phases with similar characteristic reflections, without distinguishing between individual phases of apatite.

## Data Availability

The data presented in this study are available on request from the corresponding author.

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
