# Peer review of "Quality of Phosphate Rocks from Various Deposits Used in Wet Phosphoric Acid and P-Fertilizer Production"

_materials, 2023, doi:10.3390/ma16020793_

Round 1

Reviewer 1 Report

This article provides valuable information about quality of phosphate rocks from various origins used in wet phosphoric acid and P-fertilizers production. I recommend accepting and publishing this manuscript in this current form.

Author Response

The authors would like to thank you for your thorough and substantive review of our manuscript.

Reviewer 2 Report

Introduction: At the end of the introduction, you need to write clearly the aims, objectives and relevance of the study. It is necessary to indicate why these deposits were chosen and indicate the significance of this scientific work.
243-245. Here you need to provide a link.
Discussions.
It is possible that it would be logically correct to discuss first the mineral and then the chemical composition of the raw materials.
Unfortunately, not from the introduction, not from the discussion, there is no complete impression of what this work was done for, what is new in it and what it gives for understanding the technological process. There is a similar text in the chapter "Conclusion". Perhaps it should be moved to the "Discussion", and in the "Conclusion" it is necessary to formulate the results obtained.
In general, it is difficult to understand whether the conclusions are supported by the results, since the conclusions are not formulated.

Author Response

(The authors gave the same response as above.)

Reviewer 3 Report

I have several comments about the manuscript:

1. The title contains the term “various origins” I suggest adding the origin of these phosphate rocks if there are of sedimentary or igneous origin and if there are any differences due to the origin.

2. I suggest changing the title to “Assessment of the phosphate rocks from various origins for wet phosphoric acid and P- fertilizers”.

3. Tables 2 and 4 need references.

4. Deleting the subtitle “2.2.1.1 microwave digestion” to be with the 2.2.1, because you have no 2.2.1.2.

5. Some chemical procedures need references, especially that related to wet analysis.

6. Uniform the units of Cl to be in mg.kg-1 (lines 286 and 288).

7. How the authors calculate the Cu and Zn% in lines 315 and 316.

8. Adding the permissible values to Table 5.

9. Uniform the mineral codes of the identification cards (hydroxyapatite, quartz, calcite).

10. Merging Figures 4, 5, and 6 to be one figure.

Good luck 

Author Response

(The authors gave the same response as above.)

Round 2

Reviewer 2 Report

The authors took the comments well. With all the fixes, I have no further remarks